# Role of cash transfers in mitigating food insecurity in India during the COVID-19 pandemic: a longitudinal study in the Bihar state

Sanchit Makkar,[1] Jawahar Ramasamy Manivannan,[2] Sumathi Swaminathan,[1] Sandra M Travasso,[2] Anjaly Teresa John,[2] Patrick Webb ,[3] Anura V Kurpad,[4] Tinku Thomas  [5]

¹Division of Nutrition, St John's Research Institute, Bangalore, Karnataka, India
²Division of Epidemiology and Biostatistics, St John's Research Institute, Bangalore, Karnataka, India
³Friedman School of Nutrition Science and Policy, Tufts University, Boston, Massachusetts, USA
⁴Department of Physiology, St John's Medical College, Bangalore, Karnataka, India
⁵Department of Biostatistics, St John's Medical College, Bangalore, Karnataka, India

**Correspondence to**
Dr Tinku Thomas;
tinku.sarah@sjri.res.in

## ABSTRACT

**Objective** There are scant empirical data on the impacts of the COVID-19 pandemic on food security across the globe. India is no exception, with insights into the impacts of lockdown on food insecurity now emerging. We contribute to the empirical evidence on the prevalence of food insecurity in Bihar state before and after lockdown, and whether the government's policy of cash transfer moderated negative effects of food insecurity or not.

**Design** This was a longitudinal study.

**Settings** The study was conducted in Gaya and Nalanda district of Bihar state in India from December 2019 to September 2020.

**Participants** A total of 1797 households were surveyed in survey 1, and about 52% (n=939) were followed up in survey 2. Valid data for 859 households were considered for the analysis.

**Main outcome measures** Using the Food Insecurity Experience Scale, we found that household conditions were compared before and after lockdown. The effect of cash transfers was examined in a quasi-experimental method using a longitudinal study design. Logistic regression and propensity score adjusted analyses were used to identify factors associated with food insecurity.

**Results** Household food insecurity worsened considerably during lockdown, rising from 20% (95% CI 17.4 to 22.8) to 47% (95% CI 43.8 to 50.4) at the sample mean. Households experiencing negative income shocks were more likely to have been food insecure before the lockdown (adjusted OR 6.4, 95% CI 4.9 to 8.3). However, households that received cash transfers had lower odds of being food insecure once the lockdown was lifted (adjusted OR 0.75, 95% CI 0.56 to 0.99).

**Conclusion** These findings provide evidence on how the swift economic response to the pandemic crises using targeted income transfers was relatively successful in mitigating potentially deep impacts of food insecurity.

## INTRODUCTION

The COVID-19 pandemic is known to have had serious negative effects across the world, with particularly concerning implications for poverty, malnutrition and food insecurity in low and middle-income countries (LMICs).

## STRENGTHS AND LIMITATIONS OF THIS STUDY

⇒ We studied the effect of COVID-19 pandemic on household food insecurity and the role of cash transfers in mitigating this effect, if any, in the Bihar state of India.

⇒ We used Rasch modelling to statistically validate the food insecurity information collected through longitudinal survey.

⇒ The role of cash transfers was analysed in a longitudinal study design using quasi-experimental approach after addressing for endogeneity.

⇒ The follow-up survey conducted telephonically led to dropout of some households and potentially affected the way enumerators collected information.

However, empirical evidence of such impacts remain scarce.[1] Food insecurity is defined as 'when a person lacks regular access to enough safe and nutritious food for normal growth and development and an active and healthy life. This may be due to unavailability of food and/or lack of resources to obtain food. Food insecurity can be experienced at different levels of severity'.[2] It is estimated that in South and Southeast Asia, around 33 million people may have been pushed into a state of acute food insecurity since February 2020,[3] and that between 88 and 115 million people globally were pushed into extreme poverty during 2020.[4]

While governments scrambled to curb the spread of the disease, one of the most common policy measures was to prevent the movement and congregation of individuals by imposing lockdowns that were intended to isolate vulnerable people and prevent transmission. An unfortunate (but foreseen) side effect of lockdown was increased poverty and constrained access to food due to an inability to leave the home and because of loss of jobs and income. In India, a rigorous lockdown

imposed on 23rd March 2020 brought most economic activities to a halt. The abrupt lockdown caused disruptions to food supplies, restricted labour availability and severely cut incomes, resulting in an economic downturn that affected most of India's 1.3 billion population.[5] A survey by the Centre for Monitoring Indian Economy showed a steep rise in unemployment rate across India from 8.4% in March 2020 to 23.5% in April 2020, with 75% (91 million) attributed to job loss among small traders and casual labour.[6 7] During the period of lockdown, the prices for non-cereal nutrient-dense foods (pulses, vegetables and eggs) rose faster and higher than the price of cereal foods like wheat and rice.[8] In addition, the government's long-standing supplementary feeding programmes, specifically the Integrated Child Development Services and the Mid-day Meal Programme, were suspended, which negatively impacted food consumption of entitled children and women. In an attempt to mitigate the expected adverse effects of the lockdown, the government implemented an urgent relief package valued at rupees 1.7 lakh crore (~US$22.6 billion), including both cash and food support.[9] This paper focuses on the cash transfers and its effect on food insecurity.

Cash transfers can play a significant role in diversifying the diet and improving household food consumption[10 11] because it can smooth the variability of food consumption by stabilising income, thereby protecting food intake at household level.[12] However, the impact of cash transfer initiative specific to COVID-19 has not been studied so far. Intuitively, financial assistance provided in response to COVID-19 through cash transfer instruments would have protected household purchasing power in the face of price hikes. Therefore, cash transfers would have buffered against rising food insecurity at the household level. In this context, our specific objective was to document the state of food insecurity in two districts of Bihar and the role (if any) of cash transfers in mitigating negative impacts on food insecurity at the household level.

## METHODOLOGY
### Sources of data
We conducted a longitudinal study in Gaya and Nalanda districts in Bihar. A door-to-door survey was conducted from July 2019 through September 2019, and a follow-up (denoted as survey 1) survey was undertaken from December 2019 to February 2020 (n=1732 households). Of 1797 households contacted, valid data were available for 1732. The original intent was to examine linkages between agriculture and nutrition in rural areas, with a focus on five nutrient-dense foods, namely, pulses (red lentils), milk, green leafy vegetables, eggs and poultry. A total of 142 villages in Gaya district and 134 in Nalanda district were sampled through multistage cluster sampling, and households were chosen by the random walk method (10 households per village). To account for heterogeneity in distances to district market centres, the villages were sampled from 0 to 5 km, 6–15 km, 16–30 km

and >30 km distance bands from the district headquarters. Within each village, households involved in the production of the nutrient-dense foods, landholding households (regardless of their output) and landless households were included in the sampling frame. Data on household consumption of food in the previous month were collected from all households, and information related to food production was collected from those households involved in the production of nutrient-dense foods.

When lockdown was eased, a telephonic survey (survey 2) was conducted in the same households during August and September 2020. Using the contact numbers (1797 households) provided during survey 1, all respondents were contacted telephonically. After multiple attempts, response rate was 52.3% (n=939) with >50% response rate in each district. The aim of the telephonic survey was to identify the effects of the pandemic on food production and consumption in the context of restrictions. All interviews were audio recorded, as well as documented in interviewer notes. The head of the household or his/her spouse were the respondents and the interviews were in Hindi.

The survey instrument included questions on the effects of COVID-19 on participant income/livelihoods, effects on food production and sale (among food producing households), support received in cash and in kind from the government, non-governmental organisations (NGOs) and others during the period, and effects on food purchase and consumption (among all respondents). The widely validated Food Insecurity Experience Scale (FIES), which determines food access at different levels of severity, was also administered. Complete FIES data were available from 1713 households in survey 1, and 859 in survey 2 were considered for the analysis. Figure 1 provides the framework, laying out the data sources used and the methodology adopted for the study.

### Food Insecurity Experience Scale
The household version of the FIES survey was administered to all sampled households in surveys 1 and 2. The FIES has a set of eight questions with dichotomous responses (yes or no). Specifically, to determine access to food over the prior month, questions whether the household (1) worried about not having enough food, (2) was unable to eat healthy and nutritious food, (3) ate only few kinds of foods, (4) skipped a meal, (5) ate less, (6) ran out of food, (7) was hungry but did not eat and (8) went without eating for a whole day[13] were asked. A Rasch model was used to assess the appropriateness of the FIES data collected in the context of the current study, as recommended by the Food and Agriculture Organization.[13] The model was used to statistically validate the quality of the data based on assumptions such as (1) more severe items are less frequently affirmed and (2) household that affirm a particular item on the scale are likely to affirm lesser severe items on the scale. Although these are not necessary conditions, testing of these served as a validation for the item scoring. The Rasch model is a single-parameter logistic item response theory model and

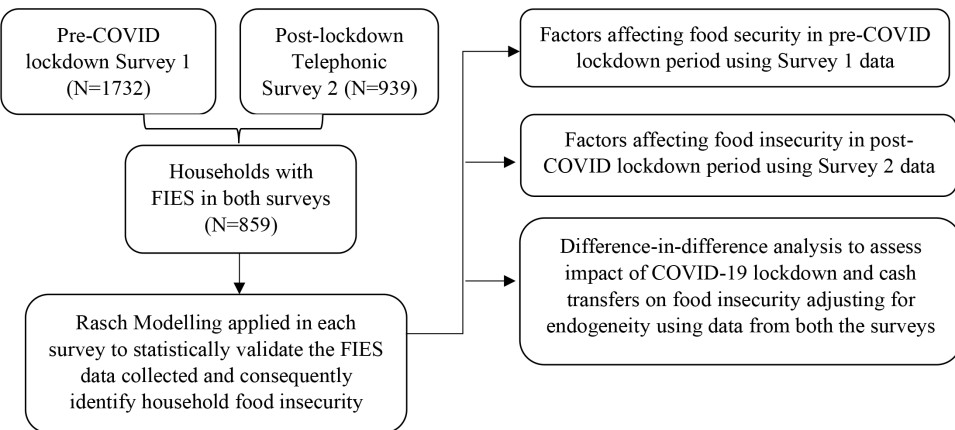

**Figure 1** Framework outlay for the data sources used and the methodology adopted. FIES is a survey which has a set of eight questions with dichotomous responses (yes or no). FIES, Food Insecurity Experience Scale.

was used to validate FIES based on these assumptions, building on the precursor Household Food Insecurity and Access Scale.[14] The domains identified as significant components of food insecurity are comparable across ethnographic groups[15] and therefore establish the appropriateness of the tool for the setting.

The four different types of results generated from the Rasch model, which are used to check the assumptions made, are fit statistics which are infit and outfit, residual correlation matrix and Rasch reliability criteria. Infit statistic measures the discriminatory power of an item, that is, items that did not perform well within the given data. The outfit statistic is like infit, but it is sensitive to responses which are extremely unexpected or outliers in the response pattern. To validate the data, a set of criteria for these measures was followed.[13 16] The adequate fit value for both fit statistics had to lie between 0.7 to 1.3. If an outfit value was >2, it was considered high. However, if infit statistics were well within range then high outfit values were not considered for eliminating the items. The residual correlation matrix for each pair of items indicates pairs that are redundant, and an absolute value of >0.4 was considered high, indicating multidimensionality between a pair of items. The last assumption (ie, Rasch reliability) accounts for the total variance explained by the model. For eight items FIES scale, a value greater than 0.7 is considered reliable and adequate.

Following statistical validation, the households having a raw score ≤3[16] were identified to be 'food secure'. This category would also include mild food insecure households. The households were categorised to be food insecure for a raw score of >3. This binary classification of food insecurity status was used as the outcome variable in the entire analysis.

## Statistical methods
Household factors associated with food insecurity were identified through logistic regressions using the survey 1 data on 1713 households. The effect of cash transfers on food insecurity was examined in survey 2 (n=859) data using logistic regression after adjusting for all household factors identified

to be associated with food insecurity in survey 1 analysis. However, cash transfers were based on an eligibility criterion, which implies a non-randomisation in sample selection. Therefore, it may cause endogeneity that could arise due to household characteristics being conducive to receiving cash benefit and hence a potential selection bias leading to biased estimate and SEs with narrower CIs.

Further, as a sensitivity analysis to account for potential endogeneity, we computed the propensity score for receipt of cash benefit considering household characteristics that were either associated with cash benefit or with household food insecurity. The variables considered for the computation of the propensity score were having a ration card or beneficiary of a public distribution system (PDS) that provides food and non-food items through fair price shops at a subsidised rates (yes vs no), household size, land owned (acres), education status (literate vs illiterate) and age of the household head, type of economic activity household is involved (casual labour, regular salaried, self-employed in agriculture and non-agriculture) and wealth quintiles from monthly per capita expenditure.

The difference in change in the odds of being food insecure between survey 1 and survey 2 (for households that received and did not receive cash benefits) was examined as an interaction effect between time (prelockdown survey and post-lockdown survey) and receipt of cash transfer, in a random effects model with logit link function, using the propensity score as a covariate. The following generalised estimating linear model was used:

$$\text{Logit}(p_{ij}) = \beta_0 + \beta_1 C_{ij} + \beta_2 T_{ij} + \beta_3 (C_{ij} \times T_{ij}) + B_4 P_{ij} + \beta_5 X_{ij} + \epsilon_{ij},$$

where $p_{ij}$ is the probability of household food insecurity for household $i$ in time $j$, where $j$ is 0 (survey 1) and 1 (survey 2); $C_{ij}$ is an indicator variable for household $i$ having received cash benefit in time $j$; $T_{ij}$ is the time dummy being equal to 0 for the pre-lockdown period and 1 for the postlockdown period for household $i$; $\beta_1$ is the population average difference in log odds of food insecurity by cash benefit, and $\beta_2$ is the population average difference in log odds of food insecurity between the two rounds of survey; $C_{ij} \times T_{ij}$ is the interaction between the

time and receipt of cash transfer. Statistical significance of interaction effect was considered at a p value of <0.1. The coefficient $\beta_3$ gives the estimated effect of cash transfer on food insecurity over time; $P_{ij}$ is the propensity score for each household; $X_{ij}$ is a matrix containing household characteristics in time j and $\epsilon_{ij}$ is the error term.

All results are expressed as OR (95% CI). All analyses were performed using Stata V.16.1.

## Patient and public involvement
No patient was involved.

## RESULTS
### Socioeconomic characteristics of the population before lockdown
Data from 1713 households were included for this analysis (table 1). The median household size was 6 (IQR 5–8), and 11% were women-headed households. Overall, only 39% of household heads were literate, of whom 80% were male. The median area of land owned by households was 0.09 acres. About 51% of the households reported having a ration card, implying that they were beneficiaries of safety net programme, the PDS. About 25% of all households reported that they were engaged in casual labour, while about 18% had regular salaries, and 56% were self-employed in agriculture and non-agriculture economic activities.

### Statistical validation of food insecurity using Rasch modelling
The infit statistics were examined separately in the subgroup of households who participated in both surveys and therefore made up the longitudinal panel (n=859). The infit statistics were adequately within the range (ie, 0.7–1.3), suggesting that the assumption of equal discrimination is met, and all the items are related to the latent trait. However, items such as 'worried' and 'healthy' had the outfit statistic of >2 in both surveys. Since the infit statistics were within range, these items were not dropped based on the outfit measure. Hence, all the eight items were retained to calculate the prevalence of food insecurity. The overall fit of the model, that is, Rasch reliability criteria (0.74 and 0.71 for survey 1 and survey 2, respectively) was greater than 0.7, suggesting a reasonably good fit. The residual correlations between pairs of items were all <0.4 implying no conditional dependence between the pairs of items. In addition, we checked these assumptions for the entire survey 1 sample (n=1713) and similar results were obtained. The raw scores from all eight items were used to classify households as food insecure (>3) or not in the two surveys.

### Prevalence of food insecurity before lockdown and after lockdown
The prevalence of food insecurity before lockdown was similar in the subsample that participated in survey 2 (20.1%) as in the entire sample of survey 1 (21.5%) (table 2). This implies that the survey 2 sample (n=859) was representative of the population covered in survey 1 (table 2). The subsample that

**Table 1** Household characteristics in the prelockdown period (N=1713)

|  | Median (IQR)/n(%)* |
|---|---|
| Age (years) | 40 (30–58) |
| Household size | 6 (5–8) |
| Land owned (acres) | 0.09 (0.02–0.56) |
| Household head education | |
| Illiterate | 1052 (61.4) |
| Literate | 661 (38.6) |
| Household head gender | |
| Male | 1526 (89.1 |
| Female | 187 (10.9) |
| PDS beneficiary | |
| Yes | 880 (51.4) |
| No | 833 (48.6) |
| Caste category | |
| Forward | 193 (11.3) |
| Other backward classes | 980 (57.2) |
| Scheduled castes | 540 (31.5) |
| Employment category | |
| Casual labour | 437 (25.5) |
| Regular salaried | 317 (18.5) |
| Self-employed in agriculture | 767 (44.8) |
| Self-employed in non-agriculture | 192 (11.2) |
| Household wealth status† | |
| Quintile 1 | 1027 (902–1128)/348 (20.3) |
| Quintile 2 | 1429 (1318–1527)/343 (20.0) |
| Quintile 3 | 1843 (1728–1956)/340 (19.8) |
| Quintile 4 | 2408 (2257–2635)/341 (19.9) |
| Quintile 5 | 3923 (3276–5437)/341 (19.9) |

*Numbers are presented as median (IQR) for continuous measures and frequency (%) for categorical measures.
†Currency unit for median (IQR) is given in Indian rupees: ~75 rupees=US$1.
PDS, public distribution system.

participated in survey 2 and did not participate were comparable in all other household characteristics as well (online supplemental table 1), except literacy of head of household. Households with illiterate heads were less likely to participate in survey 2. The prevalence of food insecurity in the subsample from survey 1 increased significantly from 20.1% before lockdown to 47.1% after lockdown (p<0.001), underlining the immediate impact of COVID-19 on the food basket of households.

### Factors affecting food insecurity before lockdown
In the pre-lockdown survey of 1713 households (table 3), the literacy of the household head was significantly associated with food insecurity (adjusted OR 1.41, 95% CI

**Table 2** Prevalence rates of food insecurity

| | Survey 1 (N=1713)* | Survey 1 subsample (n=859)† | Survey 2 (N=859)‡ |
|---|---|---|---|
| Prevalence rate of food insecurity (95% CI) | 21.5 (19.6 to 23.4) | 20.1 (17.4 to 22.8) | 47.1 (43.8 to 50.4) |

*Survey 1: pre-lockdown survey.
†Survey 1 subsample: pre-lockdown survey subsample of households followed up in post-lockdown survey.
‡Survey 2: post-lockdown survey.

1.08 to 1.83). Households that had experienced a recent negative economic shock, such as job loss or fall in income, were 6.4 times more likely to suffer food insecurity compared with households that reported either

**Table 3** Factors affecting food security in the pre-lockdown period* (N=1713)

| Food insecurity | OR | P value | 95% CI |
|---|---|---|---|
| PDS beneficiary | | | |
| Yes† | | | |
| No | 1.04 | 0.75 | 0.80 to 1.35 |
| Land owned (acres) | 1.00 | 0.24 | 1.00 to 1.00 |
| Household size | 0.96 | 0.13 | 0.90 to 1.01 |
| Household head education status | | | |
| Literate† | | | |
| Illiterate | 1.41 | 0.01 | 1.08 to 1.83 |
| Age (years) | 0.99 | 0.18 | 0.98 to 1.00 |
| Income shock | | | |
| Positive or none†§ | | | |
| Negative | 6.41 | <0.001 | 4.97 to 8.27 |
| Employment category | | | |
| Casual labour† | | | |
| Regular salaried | 0.65 | 0.02 | 0.45 to 0.94 |
| Self-employed in agriculture | 0.69 | 0.02 | 0.51 to 0.95 |
| Self-employed in non-agriculture | 0.77 | 0.26 | 0.49 to 1.21 |
| MPCE categories‡ | | | |
| Quintile 1† | | | |
| Quintile 2 | 0.89 | 0.55 | 0.61 to 1.31 |
| Quintile 3 | 0.85 | 0.40 | 0.58 to 1.24 |
| Quintile 4 | 0.63 | 0.02 | 0.42 to 0.94 |
| Quintile 5 | 0.66 | 0.05 | 0.43 to 1.00 |

*Analysis using logistic regression.
†Reference category.
‡Monthly per capita expenditure (MPCE) quintiles as a proxy for income quintile.
§Positive or no economic shocks includes found a job, hike in salary and received food and money as gift, whereas negative economic shock includes business closures, mass layoffs, price increase of commodities, job loss, wage cuts, loss of remittances, low rate for produce to be sold at market price, indebtedness and crop failure.
PDS, public distribution system.

no negative or positive economic shocks, including the finding of a new job (adjusted OR 6.41, 95% CI 4.97 to 8.27). PDS beneficiary status was not associated with food insecurity status. People engaged in a regular salaried profession or self-employment in agriculture had lower odds of experiencing food insecurity (adjusted OR 0.65, 95% CI 0.45 to 0.94, and adjusted OR 0.69, 95% CI 0.5 to 0.95, respectively) compared with casual labourers. Belonging to a higher income class, that is, top 40% (quintile 4 and quintile 5), had lower odds of being food insecure as compared with bottom 20% income class (adjusted OR 0.63, 95% CI 0.42 to 0.94, and adjusted OR 0.66, 95% CI 0.43 to 1.00, respectively).

**Factors affecting food insecurity over time, including cash transfers**

In the 859 households surveyed in the postlockdown period, about 42% households had received a cash transfer from the government. The total amount received during the three lockdown months (April–June 2020) ranged from 200 rupees to 7500 rupees, with households reporting the receipt of a median 1500 rupees (IQR 1000–1600 rupees). In addition, 55% of the households also received food as well as other in-kind benefits from various NGOs, and 28% received both cash and in-kind benefits.

Households that received cash transfers had a lower proportion of food insecurity (43.6%) compared with those who did not (49.7%). Among factors that influenced food security in the postlockdown period (table 4), households that received the transfers were 25% less likely to be food insecure compared with those that did not (adjusted OR 0.75, 95% CI 0.56 to 0.99). For other factors such as type of employment, wealth quintiles were associated with food insecurity in the same manner as the prelockdown period. The lockdown-related factors reported by households that could have contributed to food insecurity were restrictions in agricultural activities (42.3%), increased purchase prices (22%–53% depending on the type of food) and negative effect on livelihood (79.3%).

To arrive at better estimates of the effects of cash transfer on food insecurity, longitudinal data (pre-lockdown and post-lockdown surveys) of 859 household were considered. The households that received cash transfer were compared with those who did not. The proportion of PDS beneficiary households were 58% in the group that received the cash transfer (online supplemental table 2) compared with 47% in the group that did not receive this. All other characteristics were comparable between the groups. Therefore, cash benefits could be conditional

**Table 4** Factors affecting food security in post-lockdown period* (N=859)

| Food insecure | OR | P value | 95% CI |
|---|---|---|---|
| Received cash benefit | | | |
| No† | | | |
| Yes | 0.75 | 0.047 | 0.56 to 0.99 |
| PDS beneficiary | | | |
| Yes† | | | |
| No | 1.05 | 0.75 | 0.79 to 1.39 |
| Land owned (acres) | 0.91 | 0.16 | 0.81 to 1.04 |
| Household size | 1.002 | 0.94 | 0.95 to 1.05 |
| Household head education status | | | |
| Literate† | | | |
| Illiterate | 1.47 | 0.01 | 1.09 to 1.99 |
| Age (years) | 0.99 | 0.03 | 0.98 to 1.00 |
| Employment category | | | |
| Casual labour† | | | |
| Regular salaried | 0.68 | 0.08 | 0.44 to 1.05 |
| Self-employed in agriculture | 0.79 | 0.20 | 0.55 to 1.13 |
| Self-employed in non-agriculture | 0.68 | 0.12 | 0.41 to 1.11 |
| MPCE categories‡ | | | |
| Quintile 1† | | | |
| Quintile 2 | 0.90 | 0.63 | 0.58 to 1.40 |
| Quintile 3 | 0.87 | 0.53 | 0.55 to 1.36 |
| Quintile 4 | 1.00 | 1.00 | 0.64 to 1.56 |
| Quintile 5 | 0.67 | 0.09 | 0.42 to 1.07 |

*Analysis using logistic regression.
†Reference category.
‡Monthly per capita expenditure (MPCE) quintiles as a proxy for income quintile.
PDS, public distribution system.

**Table 5** Cash transfer and food insecurity for the longitudinal data* (N=859)

| Food insecurity | OR | P value | 95% CI |
|---|---|---|---|
| Time of assessment | | | |
| Pre-lockdown† | | | |
| Post-lockdown | 4.49 | <0.001 | 3.41 to 5.92 |
| Received cash benefits | | | |
| No† | | | |
| Yes | 1.15 | 0.43 | 0.81 to 1.61 |
| Time period×received cash benefits | 0.65 | 0.06 | 0.42 to 1.01 |
| PDS beneficiary | | | |
| Yes† | | | |
| No | 1.38 | 0.41 | 0.64 to 2.98 |
| Land owned (acres) | 0.91 | 0.57 | 0.67 to 1.25 |
| Household size | 0.98 | 0.41 | 0.93 to 1.03 |
| Household head education status | | | |
| Literate† | | | |
| Illiterate | 1.37 | 0.04 | 1.02 to 1.85 |
| Age (years) | 0.99 | 0.04 | 0.98 to 1.00 |
| Employment category | | | |
| Casual labour† | | | |
| Regular salaried | 0.67 | 0.02 | 0.48 to 0.94 |
| Self-employed in agriculture | 0.70 | 0.02 | 0.53 to 0.94 |
| Self-employed in non-agriculture | 0.64 | 0.02 | 0.43 to 0.94 |
| MPCE categories‡ | | | |
| Quintile 1† | | | |
| Quintile 2 | 0.85 | 0.35 | 0.60 to 1.19 |
| Quintile 3 | 0.91 | 0.59 | 0.64 to 1.29 |
| Quintile 4 | 0.89 | 0.51 | 0.62 to 1.27 |
| Quintile 5 | 0.75 | 0.13 | 0.51 to 1.09 |
| Propensity score | 5.93 | 0.62 | 0.01 to 6794.14 |

*Analysis using generalised estimation equation model.
†Reference category.
‡Monthly per capita expenditure (MPCE) quintiles as a proxy for income quintile.
PDS, public distribution system.

on eligibility criteria, which we could not assess directly. Hence, we also corrected for endogeneity arising due to selection bias by propensity scores adjusted analysis (table 5). The interaction effect of time (pre-lockdown vs post-lockdown) and receipt of cash transfer (received cash benefits or not received) was examined to quantify the differential effect of cash benefit on household food insecurity change due to the lockdown. Overall, there was a significant increase in household food insecurity in the post-lockdown survey compared with the pre-lockdown survey (adjusted OR 4.5, 95% CI 3.4 to 5.93). The interaction of time and cash benefit receipt was significant (adjusted OR 0.65, 95% CI 0.42 to 1.01; p=0.06) such that the odds of becoming food insecure in the lockdown period by the group that received cash benefits in this period was 0.65 times compared with the households that did not receive cash benefits. Hence, this group was 35% less likely to be food insecure. The adjusted OR was

0.65 (95% CI 0.44 to 0.97, p=0.03) after accounting for clustering effects of village. This underscored a beneficial effect of cash transfer on food insecurity even after adjusting for differences in familial characteristics between the households that received and did not receive cash benefits.

## DISCUSSION

This study contributes evidence on the impact of the COVID-19 pandemic-related lockdown on food security

in Bihar state, in two ways. First, it examines the status of food insecurity in the pre-lockdown and post-lockdown periods. Second, it examines the impact of cash transfers during this period on post-lockdown food insecurity. Our longitudinal survey findings suggest that there was a significant increase in the prevalence of food insecurity immediately after lockdown, such that this increased steeply, from 20% (before lockdown) to 47% (after lockdown). We also found that the social safety net programmes such as cash transfers introduced under the Pradhan Mantri Garib Kalyan Yojana, had the potential to address this stressful situation, such that the odds of being food insecure in a given household sample were much lower in comparison to households that did not receive the cash transfer. The transfers may have acted as an added income for the households, such that their purchasing power to spend on essential items did not reduce. The paper also examines how the cash transfer mechanism was beneficial to some degree to tackle this crisis.

Intervention strategies focused on reducing poverty and food insecurity in the context of COVID-19 have been widely implemented across LMICs, but not really assessed in terms of their impacts.[17 18] According to World Bank study, a total of 277 cash transfer programmes in 131 countries were introduced to handle the COVID-19 crisis by June 2020.[19] For example, India announced a relief package in March 2020 as an immediate response to the economic crisis associated with its policy decision to lockdown. Although these existing schemes were important, they did not ensure protection of the various dimensions that affect livelihoods and consequently, food affordability.

The supplementary nutrition such as Integrated Child Development Services and the Mid-day Meal Programme, were disrupted during the lockdown period and these programmes go a long way in providing quality nutrition to the beneficiary population. This study shows that a cash transfer of approximately 500 rupees/month was helpful in mitigating the negative effect of the lockdown on food insecurity and was helpful towards meeting the energy requirement of the recipient household. However, the sufficiency of this amount to meet the requirements for dietary diversity of the beneficiary households is still questionable. Some evidence suggests that cash transfers increase local food prices, steepened the price slope and potentially worsened food security for those who were ineligible for transfers,[20–22] but this was not investigated in the current study. Our research was limited to understanding the immediate impact of cash transfers on household food insecurity. In the long run, even after the lifting of the COVID-19 lockdown, the bigger problem is the need to holistically meet the nutritional requirements of the population, rather than simply their energy requirements or averting hunger. This is because the revival of the economic status of the population is likely to take longer, especially with COVID-19-related restrictions still persisting, and with repeated waves of infection.

The effect of the pandemic on food insecurity found in this study is externally valid in India, as a study in Uttar Pradesh state of India showed findings consistent with the present Bihar study, where household food insecurity increased from 21% in December 2019 to 80% August 2020, and children in these households were less likely to consume a diverse diet.[23] As far as food security among the Indian farmers is concerned, a study on tomato and wheat producers showed that the latter were relatively less affected in terms of decline in income due to fixed market prices and, hence, were less affected by food insecurity in comparison to the tomato farmers, whose income fell by almost 50%.[24] In another study, structural differences in market infrastructure in different states of India were found to be the reason for different challenges being suffered by farmers across India. For instance, in Haryana state of India, the procurement of crops were at a relatively stable prices as compared with Odisha state of India with improper procurement systems.[25]

Our findings are also consistent with experiences from other LMICs. In Bangladesh, about 90% of surveyed households experienced negative income shocks due to their lockdown, and about 88% of the households became food insecure.[26] A study in two East African countries, Kenya and Uganda, reported that during the first wave of COVID-19 pandemic, the proportion of the food insecure population increased by 38% and 42%, respectively, and more than two-thirds of the population suffered from negative income shocks.[27] A panel data study in Nigeria found an increase of 6–15 percentage points in households experiencing food insecurity when their lockdown was implemented.[28] In Nepal, food insecurity was a serious concern among the low-income and disadvantaged families during the COVID-19, and it severely affected their health and well-being.[29]

In this context, our results suggested that the size and impact of negative economic shocks were largest in comparison to other factors affecting food insecurity during the pre-lockdown period. This implies the sensitive nature of food insecurity to economic shocks and intuitively suggests that economic shocks that put stress on livelihoods would have amplified the detrimental effect on food security during the pandemic times as well. To financially cope up with this potential amplified effect of shocks, we found that households adopted strategies such as borrowing either through loan (50%) or from neighbours and relatives (48%) followed by using personal savings (46%). Other strategies adopted were cutting down on the food consumption (26.3%) and discontinuing regular medications or health check-ups (21.5%). All these strategies would probably have long-term repercussions in terms of poor diet quality. Future studies focusing on these aspects should also take these pathways into account.

## Limitations, strengths and policy implications

There are some limitations of this study. First, survey 2 was telephonic, which could have affected the way the

enumerators interpreted responses. Second, all households from survey 1 could not be followed up, probably because participants have changed their mobile phone numbers or have migrated to other areas where connectivity may be an issue, but the followed-up household was a representative subsample of survey 1.

This study contributes to the documentation and an understanding of the tangible impact of an exogenous economic shock like the COVID-19 lockdown on food insecurity and the role of social safety net programmes, such as direct benefit cash transfers in providing a shield to cope with distress situations linked to food and nutrition insecurity in rural India. The findings suggest that effective policy responses are needed to protect food intake in the context of parallel actions that impair food access, to protect food security of the population.

**Contributors** SM and TT conceptualised the manuscript and methodology. SM, SS, SMT, ATJ and TT were responsible for conducting the survey. SM conducted the data preparation and analyses. JRM contributed to the analyses. SM wrote, reviewed and edited the original draft based on the comments from all authors. JRM, SS, SMT, ATJ, PW, AVK and TT reviewed and edited the manuscript. TT is the guarantor and was also responsible for the funding acquisition. All the authors read and approved the final manuscript.

**Funding** This work was supported by the Bill & Melinda Gates Foundation, Seattle, Washington, USA (grant number OPP1194964).

**Competing interests** None declared.

**Patient and public involvement** Patients and/or the public were not involved in the design, conduct, reporting or dissemination plans of this research.

**Patient consent for publication** Not applicable.

**Ethics approval** This study involves human participants and was approved by the institutional ethics committee (IEC reference number: 297/2018) at St. Johns Medical College and Hospital (Bengaluru, India), and permissions from the state government of Bihar were obtained. Verbal informed consent was obtained from the participants prior to data collection for the telephonic survey, while written consent was obtained for the in-person surveys prior to lockdown and before taking part in the study.

**Provenance and peer review** Not commissioned; externally peer reviewed.

**Data availability statement** Data are available upon reasonable request. All data used in the study can be shared upon reasonable request.

**ORCID iDs**
Patrick Webb http://orcid.org/0000-0002-9857-3354
Tinku Thomas http://orcid.org/0000-0002-1786-6076

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
