## [Reviewer comments · BMJ Open]

ARTICLE DETAILS

TITLE (PROVISIONAL)	The role of cash transfers in mitigating food insecurity in India during the COVID-19 pandemic: a longitudinal study in the Bihar state
AUTHORS	Makkar, Sanchit; Manivannan, Jawahar; Swaminathan, Sumathi; Travasso, Sandra; John, Anjaly; Webb, Patrick; Kurpad, Anura; Thomas, Tinku

VERSION 1 – REVIEW

REVIEWER	Jean-François Maystadt UCLouvain
REVIEW RETURNED	27-Jan-2022

GENERAL COMMENTS	The study assesses the impact of lockdown on food insecurity in Bihar State and whether the government policy of cash transfer helps to moderate the negative effects of food insecurity. The topic is of great interest and policy relevant. The paper is well-written. One challenge is acknowledged by the authors: the selection into the cash transfer programs. The authors propose a Propensity score method to deal with selection. Given the non-experimental nature of the data, I found the method satisfying. I found that the paper would contribute to the literature on cash transfer and is of course highly topical. I would like however the authors to clarify the following points: Major comments • The rate of attrition is very high: 48%. I would like to see a discussion on how it would affect the results. For example, what are the characteristics, including the food insecurity status, associated with the probability to get out of the sample after survey 1?• Errors terms are likely to be correlated within villages. It is advisable to follow the sampling design to determine the level of clustering (Abadie et al. 2017: https://www.nber.org/papers/w24003)• In the analysis, the authors interact the fact to receive cash transfer with the time indicator. I would have like to see an analysis looking at the extensive margin (the amount). In that way, the authors could quantify, how much is needed to fully compensate the detrimental effect of lockdown. Furthermore, another way to look at the importance of selection is to assess the impact of the cash amount, only selecting recipient households (or at least those who had received payments prior to the lockdown). That would be similar to a dose-response model. Is it something the authors could do? Minor comments • The introduction is very clear. FYI, the World Bank estimates that by June 2020, 277 cash transfers programs in 131 countries have been introduced in response to COVID-19. That might be worth to point out (in the introduction or in the discussion). Just a suggestion.
---

	https://www.poverty-action.org/blog/giving-people-cash-during-covid-look-what-were-working  • The statistical validation of the food insecurity experience scale using Rasch modelling (pages 8-9 and page 11) was well explained. But if space needs to be gained, I would place part of it in appendix. But again, just a suggestion.
--	--

REVIEWER	Dev Ram Sunuwar Nepal APF Hospital, Kathmandu, Nepal, Department of Nutrition and Dietetics
REVIEW RETURNED	08-Feb-2022

GENERAL COMMENTS	bmjopen-2021-060624: The role of cash transfers in mitigating food insecurity in India during the COVID-19 pandemic: the case of Bihar state Food insecurity is a serious social and public health problem where many people worldwide lacked access to adequate and sufficient food, which is exacerbated by the COVID-19 pandemic. At the time of the emergence of the COVID-19 pandemic, this study is conducted to examine the influence of cash transfer in reducing negative impacts of food insecurity at the household level in two districts of Bihar, India. At the time of the emergence of the COVID-19 pandemic, many people worldwide lacked access to adequate and sufficient food. The article has written intelligible fashion and is written in Standard English. The article is presented appropriately, supported by data, and draws a conclusion. It will definitely provide adequate empirical evidence to policymakers and stakeholders. However, there are minor concerns that need to be improved before the manuscript is accepted for publication. GENERAL COMMENTS  • Initially, this study was conducted to examine the relationship between agriculture and nutrition particularly production of food covering large geographical area with larger populations. Although I could not find any cash transfer program before lock down which is considered the main factors in this study. Please mention it within the manuscript. • In my opinion, after lockdown, the government also distributed the relief packages at a local level to most of the disadvantaged and poor vulnerable households. I think the research project also provided cash to the sampled population regardless of wealth status, thus in this case some received packages from both the research project and government and some are not? Which may chance to affect the result. • Why author(s) did not use the difference-in-differences approach to control for endogeneity? Which is also an innovative way to create a quasi-experimental setting. • The statistical bias in the logistic regression model may occur, how could you report the multicollinearity among the predictor variables? • I could not locate figure 1 within the manuscript. SPECIFIC COMMENTS Abstract  • Line 45-54: (Odds Ratio=6.4, 95% CI:4.9-8.3) Does it mean adjusted odds ratio? It would be better to write "adjusted OR" instead of "Odds ratio" only, which makes it clearer to the reader and make sense from the adjusted model. Background  • Overall background section seems sound, well written. However, there is a lacking of hypotheses and problem statements. Could you
---

	add more about the implication of this study that how cash transfers program play an important role to mitigate the food insecurity concern during COVID-19 pandemic among target populations? • Line 22: Please make consistent "Covid-19" with COVID-19" Methodology  • Author(s) have used rigorous and appropriate statistical analyses. • How much cash had been provided to the sampled households before and after lockdown? Please mention it in methodology section too. • Line 15: "value greater than 0.7 is considered reliable....." this sentence begs citation. • It would be better if you present the study area map for this location (Not mandatory). Results  • Well written Discussion  • This paper may also help to discuss the insight of the food insecurity situation where this study explored food insecurity among the disadvantaged communities and low-income families during the COVID-19 pandemic in Province-2 of Nepal bordering Bihar, India. https://journals.plos.org/plosone/article?id=10.1371/journal.pone.0254954
--	--

VERSION 1 – AUTHOR RESPONSE

Reviewer: 1

Dr. Jean-François Maystadt, UCLouvain

Comments to the Author:

The study assesses the impact of lockdown on food insecurity in Bihar State and whether the government policy of cash transfer helps to moderate the negative effects of food insecurity. The topic is of great interest and policy relevant. The paper is well-written. One challenge is acknowledged by the authors: the selection into the cash transfer programs. The authors propose a Propensity score method to deal with selection. Given the non-experimental nature of the data, I found the method satisfying. I found that the paper would contribute to the literature on cash transfer and is of course highly topical. I would like however the authors to clarify the following points:

Response: *We thank the reviewer for acknowledging an important economic policy implemented in response to COVID-19 pandemic and appreciating our approach adopted in the study. Our responses to the specific points are mentioned below.*

Major comments

- The rate of attrition is very high: 48%. I would like to see a discussion on how it would affect the results. For example, what are the characteristics, including the food insecurity status, associated with the probability to get out of the sample after survey 1?

Response: *The subsample that participated in the Covid round of the survey is a representative sample of the initial sample as the food insecurity prevalence is the same (20.1%) as that in the original sample of Survey 1 (21.5%) as shown in Table 1. We have now examined if Covid survey participation is explained by any of the household characteristics using logistic regression. Although, the absence of statistical significance does not directly translate to no association, the odds ratios of all variables examine is very close to 1 which is the Null value of no association except literacy status of head of household. The households with illiterate heads were less likely to participate in survey 2.*

However, this was adjusted for in the analysis of impact of cash transfer in Table 4. The analysis of comparison of the two samples is now included as a supplementary table (Supplementary Table 1).

- Errors terms are likely to be correlated within villages. It is advisable to follow the sampling design to determine the level of clustering (Abadie et al. 2017: <https://www.nber.org/papers/w24003>)

Response: Thank you for this suggestion. We included village as cluster and obtained robust estimates of standard error in the final model. The results do not change and the estimate for interaction of time and cash transfer remains statistically significant [OR=0.65 (0.44,0.97)]. This is now added as a line in the manuscript “The adjusted OR was 0.65, 95% CI: 0.44,0.97, p=0.03 after accounting for clustering effects of village. This underscored...”

- In the analysis, the authors interact the fact to receive cash transfer with the time indicator. I would have like to see an analysis looking at the extensive margin (the amount). In that way, the authors could quantify, how much is needed to fully compensate the detrimental effect of lockdown. Furthermore, another way to look at the importance of selection is to assess the impact of the cash amount, only selecting recipient households (or at least those who had received payments prior to the lockdown). That would be similar to a dose-response model. Is it something the authors could do?

Response: In a current scenario, it's not possible to calculate the potential amount needed to fully compensate the detrimental effect of lockdown since in status – quo we did not know their income across both the time periods. However, we divided the cash transfer amount into three categories (<1500 INR, 1500 INR and >1500 INR) as 33.4 % of the household received 1500 INR as cash transfer and that was the median value. As suggested by the reviewer, we the examined the effect of the amount received on food insecurity in the recipient household. The households that received >1500 INR had even lower odds of being food insecure (OR=0.54, 95% CI:0.31, 0.95). We have chosen not to present these results in the manuscript as this does not truly represent a dose response association.

Minor comments

- The introduction is very clear. FYI, the World Bank estimates that by June 2020, 277 cash transfers programs in 131 countries have been introduced in response to COVID-19. That might be worth to point out (in the introduction or in the discussion). Just a suggestion. <https://www.poverty-action.org/blog/giving-people-cash-during-covid-look-what-were-working>

Response: Thanks for the suggestion. We have now included the same in the second paragraph of discussion. The following text has been added - “According to World Bank study, a total of 277 cash transfers programs in 131 countries were introduced to handle COVID-19 crisis by June, 2020”.

- The statistical validation of the food insecurity experience scale using Rasch modelling (pages 8-9 and page 11) was well explained. But if space needs to be gained, I would place part of it in appendix. But again, just a suggestion.

Response: We prefer to retain statistical validation in the manuscript. as it is a standard approach developed by Food and Agriculture Organization (FAO) to validate data collected for the Food Insecurity Expenditure Scale (FIES). It would also be convenient for a reader to understand the rationale behind using this model.

Reviewer: 2

Mr. Dev Ram Sunuwar, Nepal APF Hospital, Kathmandu, Nepal, Asian College for Advance Studies

Comments to the Author:

bmjopen-2021-060624: The role of cash transfers in mitigating food insecurity in India during the COVID-19 pandemic: the case of Bihar state

Food insecurity is a serious social and public health problem where many people worldwide lacked access to adequate and sufficient food, which is exacerbated by the COVID-19 pandemic. At the time of the emergence of the COVID-19 pandemic, this study is conducted to examine the influence of cash transfer in reducing negative impacts of food insecurity at the household level in two districts of Bihar, India. At the time of the emergence of the COVID-19 pandemic, many people worldwide lacked access to adequate and sufficient food. The article has written intelligible fashion and is written in Standard English. The article is presented appropriately, supported by data, and draws a conclusion. It will definitely provide adequate empirical evidence to policymakers and stakeholders. However, there are minor concerns that need to be improved before the manuscript is accepted for publication.

Response: *We are thankful to the reviewer for acknowledging the food insecurity as a significant aspect of public health problem and importance of this piece in the empirical evidence for framing policies related to food insecurity. Our responses to the specific comments are mentioned below,*

GENERAL COMMENTS

- Initially, this study was conducted to examine the relationship between agriculture and nutrition particularly production of food covering large geographical area with larger populations. Although I could not find any cash transfer program before lock down which is considered the main factors in this study. Please mention it within the manuscript.

Response: *The study did not conduct any cash transfer program. Yes, the original intent of the study was to examine the linkages between agriculture and nutrition. However, during the follow-up survey (survey 2) we also collected data on the relief packages, provided by the government and other agencies, in response to the COVID-19 pandemic. This has already been detailed in the sources of data under the methodology section in the manuscript. In the pre-COVID survey, we have identified factors associated with household food insecurity which served as a baseline and using the longitudinal data of the two surveys we examined the, effect of cash transfer on food insecurity.*

- In my opinion, after lockdown, the government also distributed the relief packages at a local level to most of the disadvantaged and poor vulnerable households. I think the research project also provided cash to the sampled population regardless of wealth status, thus in this case some received packages from both the research project and government and some are not? Which may chance to affect the result.

Response: *The research project did not provide either cash nor in-kind benefits to the participants. The study focused on understanding whether the cash transfers scheme implemented by the government in response to COVID-19 were helpful in mitigating food insecurity or not.*

- Why author(s) did not use the difference-in-differences approach to control for endogeneity? Which is also an innovative way to create a quasi-experimental setting.

Response: *Thanks for pointing this out. Because of the binary nature of dependent variable i.e. food insecurity, we used the generalized estimating linear model approach and controlled for endogeneity. We examined the interaction effect of round and cash transfer which is similar to using difference-in-differences approach.*

- The statistical bias in the logistic regression model may occur, how could you report the multicollinearity among the predictor variables?

Response: *We examined the association between the predictor variables and used them in the final multiple variable regression model after ensuring that the variables were not colinear.*

- I could not locate figure 1 within the manuscript.

Response: *As per the journal requirements, figure 1 has to be uploaded in a separate file and only the figure legend along with the text of the manuscript after the references section.*

SPECIFIC COMMENTS

Abstract

- Line 45-54: (Odds Ratio=6.4, 95% CI:4.9-8.3) Does it mean adjusted odds ratio? It would be better to write "adjusted OR" instead of "Odds ratio" only, which makes it clearer to the reader and make sense from the adjusted model.

Response: *The odds ratio has been indicated as adjusted OR as suggested by the reviewer.*

Background

- Overall background section seems sound, well written. However, there is a lacking of hypotheses and problem statements. Could you add more about the implication of this study that how cash transfers program play an important role to mitigate the food insecurity concern during COVID-19 pandemic among target populations?

Response: *Thanks for the suggestion. We have added the text on the same in the last paragraph of introduction section as "Intuitively, financial assistance provided in response to COVID-19 through cash transfer instruments would have protected household purchasing power in the face of price hikes. Therefore, cash transfers would have buffered against rising food insecurity at the household level. In this context,"*

- Line 22: Please make consistent "Covid-19" with COVID-19"

Response: *Thanks for pointing this out. We have now replaced "Covid -19" with "COVID -19" wherever applicable in the manuscript.*

Methodology

- Author(s) have used rigorous and appropriate statistical analyses.
- How much cash had been provided to the sampled households before and after lockdown? Please mention it in methodology section too.

Response: *Before lockdown the cash was not provided to the sampled households. After lockdown following the announcement of relief package by the government the cash benefits were provided by the government to the households. The summary on the cash benefits post lockdown has already been added as a text in the "Factors affecting food insecurity over time, including cash transfers" heading under the results section.*

- Line 15: "value greater than 0.7 is considered reliable....." this sentence begs citation.

Response: *The references for the validation criteria has already been mentioned in the same paragraph in the following line as "To validate the data, a set of criteria for these measures were followed."*

- It would be better if you present the study area map for this location (Not mandatory).

Response: *The study was done in two districts of Bihar state which is not representative for the country. Hence the authors feel that including the location map would not improve the interpretability of the paper.*

Results

- Well written

Discussion

- This paper may also help to discuss the insight of the food insecurity situation where this study explored food insecurity among the disadvantaged communities and low-income families during the COVID-19 pandemic in Province-2 of Nepal bordering Bihar, India.

<https://journals.plos.org/plosone/article?id=10.1371/journal.pone.0254954>

Response: *Thanks for highlighting this paper. We have now quoted the same in the discussion section as "In Nepal, food insecurity was a serious concern among the low-income and disadvantaged families during the COVID-19, and it severely affected their health and wellbeing."*

VERSION 2 – REVIEW

REVIEWER	Jean-François Maystadt UCLouvain
REVIEW RETURNED	28-Apr-2022

GENERAL COMMENTS	I am fully satisfied with the changes made to the manuscript and the answers to my comments (beyond expectations). Let me stress again that I believe the topic is of great interest and policy relevant. I strongly recommend to accept the paper for publication.
--

REVIEWER	Dev Ram Sunuwar Nepal APF Hospital, Kathmandu, Nepal, Department of Nutrition and Dietetics
REVIEW RETURNED	10-May-2022

GENERAL COMMENTS	The authors have addressed my all comments and concern. Therefore, this paper is now suitable for publication.
---